# The Assembly of Super-Complexes in the Plant Chloroplast

**DOI:** 10.3390/biom11121839

**Published:** 2021-12-07

**Authors:** Kezhen Qin, Alisdair R. Fernie, Youjun Zhang

**Affiliations:** 1Max-Planck-Institut für Molekulare Pflanzenphysiologie, Am Mühlenberg 1, 14476 Potsdam-Golm, Germany; K.Qin@mpimp-golm.mpg.de; 2Center of Plant Systems Biology and Biotechnology, 4000 Plovdiv, Bulgaria

**Keywords:** chloroplast, super-complexes structure, photosystem, ATPase, electron transport chain, Calvin–Benson cycle, *de novo* purine biosynthesis

## Abstract

Increasing evidence has revealed that the enzymes of several biological pathways assemble into larger supramolecular structures called super-complexes. Indeed, those such as association of the mitochondrial respiratory chain complexes play an essential role in respiratory activity and promote metabolic fitness. Dynamically assembled super-complexes are able to alternate between participating in large complexes and existing in a free state. However, the functional significance of the super-complexes is not entirely clear. It has been proposed that the organization of respiratory enzymes into super-complexes could reduce oxidative damage and increase metabolism efficiency. There are several protein complexes that have been revealed in the plant chloroplast, yet little research has been focused on the formation of super-complexes in this organelle. The photosystem I and light-harvesting complex I super-complex’s structure suggests that energy absorbed by light-harvesting complex I could be efficiently transferred to the PSI core by avoiding concentration quenching. Here, we will discuss in detail core complexes of photosystem I and II, the chloroplast ATPase the chloroplast electron transport chain, the Calvin–Benson cycle and a plastid localized purinosome. In addition, we will also describe the methods to identify these complexes.

## 1. Introduction

Protein super-complexes are composed of several protein complexes which assemble into a supramolecular structure in order to execute their biological functional such as the well-documented mitochondrial electron transport chain (mETC) complexes containing multi-subunit protein complexes [1]. The association of mETC proton translocating complexes (complex I, III and IV) as well as the mobile electron carriers (ubiquinone and cytochrome c) was revealed by protein structure studies and suggested to possibly mediate the substrate channeling of proton translocation in order to generate a proton-motive force for ATP synthesis [1,2,3]. Although the functional significance of full respirasomes is not fully understood, it has been suggested that the association of respiratory enzymes into super-complexes may increase metabolism efficiency and reduce oxidative damage [4]. It has been demonstrated that that super-complexes comprising Complex IV had higher Complex I and III activities, indicating that the association of Complex IV modifies super-complexes to enhance catalytic activity [5] (Figure 1). Indeed, even increasing evidence has been accumulated to suggest the presence of respirasomes (the super-complex of mETC Complex I, III, and IV) is essential for the stability and function of the Complex II [6]. In addition, the dynamical assembly of super-complex has also be suggested to help the organization of electron flux to optimize the use of substrates [7]. The association of super-complexes increased electron transport by reducing the distance for diffusion of cytochrome c between the cytochrome bc1 complex and cytochrome c oxidase and thereby increases competitive fitness in yeast [8]. However, the evidence of substrate channeling in the super-complexes of mETC remains a matter of debate since it is unclear if structural evidence is consistent with the occurrence of channeling or not [4,9,10,11]. Variations between these super-complexes have been reported across kingdoms, and this super-complex is now widely used as a model for super-complex research. In plants, given that the chloroplast also provides energy continuously during light conditions, the mETC association was reported to be more divergent from other species with utilizing alternative respiratory substrates being substantially different size [12,13]. In addition, although the mechanism of the super-complex association remains unclear, several studies have suggested that the formation of mETC super-complexes may be dependent upon the lipid composition of the mitochondrial membrane, and in particular, may require the presence of cardiolipin, a unique mitochondrial lipid [14]. For example, the components of mETC super-complex was significantly reduced with lacking cardiolipin which was suggested to stabilize the super-complex formation by neutralizing the charges of lysine residues between the interaction domain of complex III and complex IV in yeast [15]. Moreover, the trimer and tetramer complex III/IV super-complex could be reconstituted from purified mETC complex from yeast and exogenous cardiolipin liposomes [16]. The presence of cardiolipin in plants has however yet to be reported.

Given that the assembly of super-complexes appears rely on dynamic, transient and protein-surface structure dependent interaction, it was also previously suggested to be a mechanism by which substrate channeling could be maintained in order to regulate the metabolic flux by association and dissociation. It is usually suggested that larger complexes organized by an indefinite number of small protein complexes from the same pathway, while a super-complex association can be more simply organized by the coming together of two complexes, or a defined number of stoichiometrically interacting complexes [17]. Thus, super-complex assembly is simply several sub-complexes of molecules held together by noncovalent bonds.

In photosynthesis, energy is transferred from the sun to metabolites by the process of a water-splitting and oxygen-evolving reaction which is catalyzed by four major protein super-complexes namely photosystem I (PSI), photosystem II (PSII), cytochrome b6f (plastoquinone-plastocyanin oxidoreductase) and F-ATPase (proton-motive force-driven ATP synthase) [18]. A multi-subunit photosynthetic super-complex is suggested to be located in the thylakoid membranes of cyanobacteria, algae, and plants. Several high-resolution structures of photosynthetic complexes help us to further understand the electron flux during photosynthesis such as the complex of PSI and the light-harvesting complex I (LHCI) the latter of which was suggested to mediated the energy transfer and photoprotection [19]. In photosynthesis, the primary photosynthetic reactions are light driven, and the harvested energy is used by PSI and PSII to transport electrons across the membranes in order to establish a membrane proton gradient. PSII comprises a series of light-harvesting complexes II (LHCIIs) to form the antenna system associated with chlorophylls, carotenoids, lipids and integral membrane proteins. These complexes have the function in capturing and transmitting light energy and also have pivotal roles in photoprotection under high-light conditions through a mechanism known as the non-photochemical quenching process [20].

In this review, we will detail not only this super-complex but other supramolecular protein assemblies of the chloroplast. We will specifically discuss the core complexes of photosystem I and II, the chloroplast ATPase the chloroplast electron transport chain, the Calvin–Benson cycle and the possibility of a plastid localized purinosome.

## 2. Molecular Organization of Different Complexes

### 2.1. Photosynthesis Complex

#### 2.1.1. Photosystem I Core Complex

Plants and algae have a monomer PSI core complex that connects with different LHCIs to produce the PSI–LHCI super-complex [21,22,23] (Figure 2A). The crystal structure of pea PSI–LHCI was originally identified in 2003, and it has subsequently been described with increasing resolutions from 4.4 to 2.6 Å [19,24,25,26,27]. A cryo-EM structure of maize PSI–LHCI–LHCII at a resolution of 3.3 Å was subsequently described [28]. Almost all Lhca/Lhcb proteins from various photosynthetic species have been described structurally. The antenna size, protein content, and related colors of the LHCI proteins varied greatly among species, according to these structural investigations. [29,30]. The structural similarities and differences among various LHCI proteins are explored in the following sections.

The PSI core of the plant is connected to four LHCI subunits, which create a semispherical belt on the PsaF/PsaJ side. Lhca1–3 are structured as Lhca1–Lhca4 and Lhca2–Lhca3 heterodimers, with Lhca3 and Lhca1 at the two edges and tightly interacting with PsaK/PsaA and PsaG/PsaB, respectively. In the crystal structures of pea PSI–LHCI, Lhca2 and Lhca4 are farther away from the core subunits, whereas Lhca2 contacts the core component PsaN at the luminal surface, as indicated in the maize PSI–LHCI–LHCII structure. Despite the fact that plants have six *lhca* genes (*lhca1–6*), only sub-stoichiometric quantities of *lhca5* and *lhca6* are expressed. [31].

Recently, stable plant PSI–LHCII super-complexes have been isolated and electron microscopy analysis show that LHCII is bound to PSI in 1:1 ratio when it is phosphorylated during state transition [32]. The protein structure of the PSI–LHCI super-complex suggests that energy absorbed by LHCI could be efficiently transferred to the PSI core by avoiding concentration quenching [19].

#### 2.1.2. PSII Core Complex

PSII is a large multi-subunit protein complex with a dimeric core and a number of membrane-attached antenna complexes on the periphery. Depending on the organism, the core complex has anywhere from 20 to 23 protein subunits. The reaction center, which is substantially conserved among plants, algae, and cyanobacteria, is the catalytic heart of the core. PsbA (D1), PsbB (CP47), PsbC (CP43), and PsbD (D2), the biggest membrane-intrinsic subunits, make up the reaction center. The photochemical reaction center, made up of PsbA and PsbD, is where charge separation and electron transport across the membrane take place. The subunits bind a total of six chlorophylls (Chls), with two pheophytins which are separately bound to PsbA and PsbD. The internal antenna proteins, PsbB and PsbC, bind multiple chlorophylls. These two subunits are responsible for light gathering and transmitting excitation energy from the antenna’s periphery to the photochemical reaction core [33]. PsbE, PsbF, PsbH, PsbI-M, PsbTc, PsbX, PsbY, and PsbZ are tiny intrinsic subunits that are found in all organisms, in addition to the reaction center. These subunits are physically and functionally preserved; however, they are less closely related between organisms than reaction center subunits. In comparison to its bacterial cousin, plant PsbY possesses one extra membrane-spanning helix. Plants lack the cyanobacterial component Ycf12 but do have the PsbTn and PsbW subunits. Their position in the plant super-complex structure might be determined [34].

The dimeric PSII core is connected with multiple LHCIIs (Lhcb proteins that bind pigments) in plants. The primary light-harvesting complex LHCII, which comprises three apoproteins (Lhcb1, Lhcb2, and Lhcb3) that form Lhcb1/Lhcb2/Lhcb3 heterotrimers in vivo with various stoichiometries [35] or in vitro with homotrimers of Lhcb1 or Lhcb2 [36,37], is the most prevalent antenna complex in vascular plants. Three monomeric minor Lhcbs are linked to the PSII core in addition to the trimeric LHCII. Based on the apparent molecular mass of their apoproteins, they were given the names CP29 (Lhcb4), CP26 (Lhcb5), and CP24 (Lhcb6) [38].

PSII–LHCII super-complexes are formed when the plant PSII core interacts with several LHCIIs. The trimeric LHCII can be characterized as strongly (S), moderately (M), or loosely (L) coupled with the PSII core (C) complex based on their respective affinities [39]. PSII super-complexes modify their antenna size in response to variations in light intensity [40,41] (Figure 2B). In plants adapted to high light conditions, the C_2_S_2_ and C_2_S types of PSII–LHCII super-complexes, which contain S–LHCII, CP29, and CP26, are the dominant species, whereas in plants adapted to low light conditions, the abundance of the larger complexes C2S2M2 and C2S2M, which bind S–LHCII, M–LHCII, and all minor antennae, increases [40,42]. The L–LHCII trimers are found on PSII’s periphery. In response to light quality changes, a fraction of L–LHCII (mobile LHCII) can be reversibly phosphorylated and migrate between PSII and PSI to balance the excitation levels of the two photosystems, a process known as state transitions [43,44]. Lhcb1 is the most abundant component in the LHCII complex, with a predicted ratio of roughly 8:3:1 for Lhcb1, Lhcb2, and Lhcb3 [45,46]. n the C_2_S_2_M_2_/C_2_S_2_M super-complexes, the exclude M–LHCIIs (Lhcb3) can also interact with CP24 [47,48]. The mobile LHCIIs can be phosphorylated in the same way that pLhcb2, which is responsible for the PSI–LHCI–LHCII super-complex formation [28,32,49,50]. In plants, the protein structures of all six Lhcb proteins were determined. At resolutions ranging from 2.5 to 2.8 Å, the crystal structures of spinach and pea LHCII (modeled as the Lhcb1 homotrimer) as well as spinach CP29 were determined [51,52,53]. A cryo-EM study of PSII–LHCII and PSI–LHCI–LHCII super-complexes at a resolution of 2.7 to 3.2 Å provided structural data for all other Lhcb proteins [28,34,48].

The peripheral antennae of PSII are structured into two layers around the core complex, according to analyses of PSII–LHCII super-complex structures [47,54,55]. The inner layer, which interacts with the dimeric core to produce the C2S2 super-complex, contains two copies of S–LHCII, CP29, and CP26. The C_2_S_2_M_2_ super-complex is formed when two CP24s and two M–LHCIIs attach to the outer layer of C_2_S_2_. S–LHCII is near to the core complex of the PSII–LHCII super-complex and connects CP26 and CP29 (from the neighboring monomer) through monomer A and B, respectively. S-third LHCII’s monomer (C) is found at the periphery, where it interacts with M–LHCII′ in the larger C_2_S_2_M_2_ super-complex. In the C_2_S_2_M_2_ super-complex, CP24 interacts extensively with CP29 and forms a CP24–CP29 heterodimer, which closely resembles the Lhca3–Lhca2 and Lhca4–Lhca1 dimers in plant PSI [19,27]. The M–LHCII trimer, which contains Lhcb3 protein, simultaneously binds to the CP24–CP29 heterodimer and S–LHCII′. Both Lhcb3 and CP24 are specific to land plants [56] and are present only in the larger PSII–LHCII super-complexes, such as C_2_S_2_M_2_ [47,48]. M–LHCII is tightly connected with CP24 in the super-complex through Lhcb3. The contacts between M–LHCII and S–LHCII or M–LHCII and CP29, on the other hand, are less extensive and mostly consist of hydrophobic interaction, independently of the enthalpy and entropy rather than intermolecular forces such as der Waals forces or hydrogen bonds. The importance of CP24 in binding M–LHCII to CP29 might explain why knocking out CP24 causes the C_2_S_2_M_2_ super-complex to dissociate, leaving only C_2_S_2_. Furthermore, when CP24 dissociates from CP29, M–LHCII may detach from the super-complex, guaranteeing that plant PSII’s antenna size varies rapidly and dynamically in response to high light. [57].

Although the high-resolution structure of the PSII–LHCII super-complex comprising L–LHCII has yet to be determined, prior negatively stained electron microscopy investigations indicated that L–LHCII interacts with the core complex near CP26 or at the periphery of S- and M-trimers [39,45,58]. Unlike S–LHCII and M–LHCII, which are usually attached to PSII, L-mobile LHCII’s LHCII can associate with PSI core under appropriate conditions. According to previous research, the mobile LHCII is mostly made up of the Lhcb1–Lhcb1–Lhcb2 trimer and may switch between PSI and PSII during state transitions. [32,50,59]. pLhcb2 is responsible for the interaction between mobile LHCII and PSI, despite the fact that Lhcb1 can also be phosphorylated. [32,50]. The entire structure of Lhcb2 and its phosphorylated Thr3 residue in the N-terminal region of maize PSI–LHCI–LHCII were first disclosed in the cryo-EM structure of maize PSI–LHCI–LHCII [28]. The PSI core subunits engage with the N-terminal region of pLhcb2, particularly the phosphate group of phosphorylated Thr3 (pThr3) and two basic residues, in the complex (Arg1 and Arg2). Furthermore, the protein sequence of Lhcb2’s N-terminal region is somewhat shorter than that of Lhcb1, suggesting that the interaction between Lhcb2 and PSI core is more specialized [28].

Plant Lhcb proteins are members of the LHC superfamily and share comparable sequences and structural characteristics. Lhcb proteins’ N- and C-termini are found on the stromal and luminal sides, respectively. Helix B, C, and A are the three transmembrane helices found in all Lhcb proteins (from the N-terminus to the C-terminus). Except for CP24, which has a very short helix E but no helix D due to its short C-terminal fragment, all Lhcbs have two amphipathic helices with identical lengths (D and E) at the luminal surfaces. In the center of the membrane, helices A and B interlock to generate a left-handed supercoil. Helix C is approximately perpendicular to the membrane plane, with a BC loop and an AC loop connecting it to helices B and A, respectively. Despite the general resemblance of their apoprotein folds, there are considerable variances in their loop regions.

In terms of sequence and structure, CP26 is the most similar to Lhcb1 of the three minor Lhcb proteins, with conformational variations being confined to their loop regions. Specifically, the N-terminal segment of CP26 adopts a shape similar to that of Lhcb1, which is consistent with a previous research indicating that CP26 may form a trimer in lieu of Lhcb1 and Lhcb2 in *Arabidopsis thaliana* when these proteins are absent. [60,61]. In the Lhcb family, the CP29 component possesses the biggest apoprotein and the longest N-terminal domain. In the PSII–LHCII super-complex, its N-terminal region forms a lengthy hairpin loop that practically covers the whole core component CP47 at the stromal surface. As a result, of all the main and minor Lhcb proteins in the super-complex, the antenna complex’s CP29 is the most intimately connected with the PSII core. The N-terminal loop of CP29 was not defined in the previously established crystal structure of CP29 due to proteolysis during crystallization. Based on these structural observations, the N-terminal portion of CP29 in the super-complex may be maintained by interactions with PSII core subunits [62], When CP29 is detached from the core, it may be quite flexible. CP24 is the smallest of the Lhcb proteins, with a short BC loop and C-terminal tail. It, on the other hand, has the longest AC loop, which contains two extra segments required for CP24–Lhcb3 and CP24–CP29 interactions. On the stromal side, CP24 has a longer helix C than the other Lhcbs, which aids in the connection with the nearby CP29 protein.

#### 2.1.3. ATPase

The F1F0-ATP synthase, also known as the ATPase synthase complex of green plant chloroplasts, is a member of the F-type ATP synthase family. Prokaryotes and eukaryotes both have mitochondria with similar sorts of complexes. Through evolution, the ATP synthase enzymes have remained impressively conserved. Bacterial enzymes are structurally and functionally similar to those found in the mitochondria of animals, plants, and fungus, as well as the chloroplasts of plants. They are always made up of three parts: a hydrophilic, almost spherical headpiece (F1) connected to a smaller membrane-bound F0 moiety by a stalk region. A central stalk connects to a peripheral stator connection in the stalk area. Three noncatalytic a-subunits and three catalytic h-subunits of around 55 kDa alternate in a hexagon to make up the majority of the bulk of the F1 headpiece. The isolated F1-ATPase produces ATP, which drives the rotation, and subunit g of 35 kDa covers most of the middle shaft. The F1F0 complex’s g subunit is linked to an 8-kDa c subunit made up of two membrane-bound a-helices arranged in a hairpin. Subunit c exists as a multimer at all times [63]. The chloroplast subunit III, the equivalent of c, forms a fixed ring of 14 subunits [64].

Rotation of the c component multimer in undamaged chloroplasts is caused by the proton motive force acting across the thylakoid membrane. This causes g to rotate, which leads to the production of ATP via rotational catalysis. A second stalk, or stator, links the F1 headpiece and F0 to prevent fruitless spinning. Green plant subunits I, II, and IV, as well as y, are engaged in the stator, and a ninth component, q, controls catalytic activity by binding to g, bringing the total number of subunits to nine. Three tiny subunits and one inhibitor protein are found in vertebrate mitochondrial ATPase.

The three catalytic h-subunits vary in conformation and bound nucleotide in the crystal structure of bovine mitochondrial F1-ATPase measured at 2.8 Å resolution [65]. In intact ATP synthase, the structure supports a catalytic mechanism in which the three catalytic subunits are in distinct phases of the catalytic cycle at any one time. Interconversion of states is accomplished by rotating the α-helical subunit’s domain relative to the a3–h3 subassembly. Based on the homologous structure of the bovine mitochondrial enzyme, the structure of the F (1)—ATPase from spinach chloroplasts was determined to 3.2 Å resolution by molecular replacement [66]. The a- and h-subunits display a structure that was quite similar to the mitochondrial and thermophilic subunits. NMR has established the structures of several minor subunits (δ, ε), but no entire ATP synthase complex has yet been crystallized. It is possible that the stator is too delicate to crystallize. As a result, we do not know the specific structure of the plant component IV (named in prokaryotes and mitochondria). Despite this, the general locations of all subunits are quite well established.

For a long time, the F-type ATP synthase was thought to be a monomeric membrane complex. The enzyme from yeast mitochondria, however, was discovered to be dimeric, utilizing the technique of blue native gel electrophoresis [67]. In comparison to the monomer, the dimer’s subunit makeup indicated the existence of three more small proteins. The creation of the dimeric state required two of these dimer-specific subunits of the ATP synthase. Because other respiratory chain complexes such as cytochrome reductase (Complex III) and cytochrome oxidase (Complex IV) also create particular types of associates, the mitochondrial ATPase dimer is not a unique large super-complex. A systematic search for large super-complexes in plant mitochondria. Three high-molecular mass complexes of 1100, 1500, and 3000 kDa were identified and separated using blue-native polyacrylamide gel electrophoresis [68]. The 1100-kDa complex was identified as dimeric ATP synthase by mass spectrometry. This dimer was only stable at extremely low detergent doses. There is no evidence, however, that the chloroplast ATP synthase forms dimers within the membrane or has particular connections with another big membrane complex.

#### 2.1.4. Cytochrome *b6/f*

The cytochrome *b6/f* complex is a dimeric integral membrane protein complex with eight-to-nine polypeptide subunits that has a molecular weight of around 220 kDa [69]. The four largest have well defined roles. Together with the 17-kDa subunit IV, which has three transmembrane helices and two b-type hemes, the 24-kDa cytochrome *b6* subunit has four transmembrane α-helices and contains two b-type hemes. The N- and C-terminal portions of cytochrome *b* of the *bc1* complex from the respiratory chain in mitochondria are homologous to cytochrome *b6* and subunit IV. The 19-kDa Rieske iron–sulfur protein, consisting of an N-terminal single transmembrane α-helix domain and a 140-residue soluble extrinsic domain with a linker region connecting these two domains, has an overall function similar to that of the iron–sulfur protein in the *bc1* complex, deprotonating the membrane-bound quinol and transferring electrons from the quinol to the membrane-bound c-type cytochrome. The 31-kDa c-type cytochrome *f* subunit is functionally related to, but structurally completely different from, the cytochrome *c1* in the *bc1* complex. In addition to these four large subunits four smaller subunits, PetG, PetL, PetM and PetN, are each bound the complex with one membrane-spanning α-helix (Figure 3B). They have no counterparts in the cytochrome bc1 complex.

X-ray structures at 3.0 Å resolution of the complex from the thermophilic cyanobacterium *Mastigocladus laminosus* [70] and at 3.1 Å resolution from the alga *C. reinhardtii* [71] have been obtained in 2003. The structure of the *b6/f* complex bears similarities to the respiratory cytochrome *bc1* complex [72] but also exhibits some unique features, such as binding one β-carotene and one chlorophyll a, and an unexpected heme sharing a quinone site. This heme is atypical as it is covalently bound by one thioether linkage and has no axial amino acid ligand. This heme may be the missing link in oxygenic photosynthesis [73]. This heme structural information suggests it is an essential factor for quinone reduction at the Q_i_ site and also facilitate rapid electron transfer and significant interactions between hemes c_i_ and b_i_. The functions of the chlorophyll and β-carotene cofactors are unknown.

The cytochrome *b6/f* complex takes electrons from PSII via plastoquinone and transfers them to PSI via reducing plastocyanin or cytochrome *c6* in the linear electron transfer scheme. This causes proton uptake from the stroma and release to the lumen, which creates a proton electrochemical gradient across the membrane, powering the Q-cycle and increasing ATP synthesis at the price of lowering equivalents. Unlike its mitochondrial and bacterial counterpart, cytochrome *bc1*, cytochrome *b6/f* can convert from linear electron transfer between both photosystems to a cyclic mode of electron transport around PSI via an unidentified mechanism. Furthermore, the cytochrome *b6/f* complex is thought to regulate state transitions by activating a protein kinase [74,75].

There is no direct evidence that the cytochrome *b6/f* complex can be associated to any other large complex of the thylakoid membranes [76], but the 35-kDa ferredoxin: -NADP^+^ oxidoreductase can bind with one copy. For ferredoxin-dependent cyclic electron transport, this connects to the main electron transfer chain [69]. It has been proposed that the cytochrome *b6/f* complex forms a super-complex with PSI, to sustain fast cyclic electron transport in the stroma lamellae [77], but there is no direct structural evidence to support this proposal. It has also been proposed that “small plastoquinone diffusion micro-domains” would exist in grana membranes [78,79], in which a few PSII complexes and a cytochrome *b6/f* complex share a domain in which plastoquinone can quickly migrate between PSII and *b6/f*. Such microdomains were proposed to be required because plastoquinone diffusion was estimated to be very slow in grana membranes [80,81]. About 70% of the PSII centers should be present in such domains, while the remaining PSII centers are present in (much) larger domains [79]. However, direct experimental evidence for the existence of such domains is currently lacking.

### 2.2. The Chloroplast Electron Transport Chain

In the primary reactions of photosynthesis, there are two modes of electron flow: linear or non-cyclic electron flow (LEF) and cyclic electron flow (CEF). In LEF, both PSI and PSII are involved, together with the cyt *b6/f* complex. LEF is more important under normal conditions than CEF, in which PSI and cyt *b6/f* are the main relevant players, without the involvement of PSII. The LEF produces both ATP and NADPH, but the CEF pathway generates extra ATP. For a balanced CO_2_ fixation and other metabolic processes LEF does not provide enough ATP (vs NADPH). By tuning the levels of LEF and CEF, the ATP/NADPH ratio can be adjusted to meet cellular demands [82,83]. Under certain conditions CEF appears to contribute substantially to photosynthetic electron flow, for instance during induction of photosynthesis and under stressful conditions such as drought, high light and extreme temperatures [82].

Currently two CEF pathways are known. They depend either on the proton gradient regulation 5 (PGR5) together with PGR5-LIKE1 (PGRL1) complex [84] or the NDH complex [85,86]. Whereas the PGR5-dependent pathway is efficient in the control of the ATP/NADPH ratio and essential to induce dissipation of energy by non-photochemical quenching, the NDH-dependent pathway is considered not to affect non-photochemical quenching. Rather it alleviates the stromal over-reduction under stress conditions and prevents photosynthesis from photoinhibition [87]. Thus, both pathways are extremely important for photoprotection and photosynthesis. However, recent analysis of high CEF arabidopsis thaliana mutants revealed a key role of the NDH complex in enhanced cyclic electron transport and augmenting production of ATP [88]. This is in line with previous observations that accumulation of hydrogen peroxide in barley under photooxidative stress was found to mediate the induction of the NDH and to enhance its activity. Clearly, the NDH-pathway of cyclic electron transport seems to be underestimated at least under environmental stress conditions, where it takes over a major role in cyclic electron transport. It appears that several types of super-complexes of PSI are involved in the regulation of LEF and CEF.

A super-complex involved in both electron transport and photosynthesis is the PSI–LHCI–LHCII–FNR–cyt *b6/f*–PGRL1 super-complex, named here the PSI-cyt *b6/f* super-complex, for the sake of simplicity. This super-complex was isolated from the green alga *C. reinhardtii* in state 2 by sucrose gradient centrifugation and has an approximate mass of 1200–1400 kDa [89]. The main components are PSI and the cyt *b6/f* complex. In addition, three smaller components are present: LHCII, ferredoxin-NADPH reductase (FNR) and PGRL1. FNR is a redox protein that can oxidize ferredoxin molecules. The function of FNR is reduction of NADP^+^ to NADPH in LEF with electrons from ferredoxin. PGRL1 was identified as a membrane protein to bind to PSI. Later, on its central role in the regulation of cyclic electron flow was established. It functions as a ferredoxin-plastoquinone reductase [90]. In a PGR5-dependent way, PGRL1 interacts with PSI and presumably also with FNR, and can take electrons from ferredoxin. PGRL1 also interacts with cyt *b6/f* and has the ability to transfer electrons to plastoquinone. [90,91]. However, this complex’s structural characterization is lacking. The most straightforward technique to determine a structural map is to photograph particles from sucrose gradient fractions using EM and do single particle image processing, which includes substantial image analysis via particle projection sorting. [92].

### 2.3. Calvin–Benson Cycle (CBC)

Oxygenic phototrophs such as cyanobacteria, algae, and land plants convert carbon dioxide and water into carbohydrates and release the by-product oxygen, a process that can be divided into the light reactions and the CBC (light-independent reactions). ATP and NADPH are produced in the light reaction process before being utilized by the enzymes of the CBC responsible for CO_2_ assimilation [93]. The CBC is regulated by light/dark transitions through the redox states of the chloroplast stroma. Chloroplast thioredoxins (TRXs), including TRX *f* and TRX *m*, are reduced by PSI upon illumination through ferredoxin and the ferredoxin thioredoxin reductase system [94,95] and further reduce and activate the enzymes of the CBC, including phosphoribulokinase (PRK) and glyceraldehyde-3-phosphate dehydrogenase (GAPDH). In addition, the small chloroplast protein CP12, which is central for the regulation of the CBC and usually possesses two Cys pairs at both N- and C-terminal regions, is redox-regulated by TRXs [33,96,97].

The GAPDH/CP12/PRK complex is formed when oxidized CP12 interacts with the GAPDH tetramer and then assembles with the oxidized PRK dimer [98,99,100,101] (Figure 4A). Both GAPDH and PRK have been found to have their enzymatic activities inhibited when they form a complex, and the activity of PRK in the complex is even lower than the free oxidized PRK dimer [102]. The activation of PRK in the complex occurs quickly, while the transition of free oxidized PRK to its active state is slower [99]. However, the mechanism for this divergence remains unknown. According to a later study, the GAPDH/CP12/PRK complex has a pool of both enzymes that is ready to be released with full activity [102]. CP12 has been hypothesized to have a variety of activities, including protecting both GAPDH and PRK from oxidative stress, in addition to regulating the CBB cycle [103] and to facilitate the stabilization of PRK during or after its synthesis in vivo [104].

Several models of the GAPDH/CP12/PRK complex were previously proposed based on small-angle X-ray scattering analysis [105]. Recently, a cryo-electron microscopy (cryo-EM) structure of the cyanobacterial GAPDH/CP12/PRK complex from *Thermosynechococcus elongatus* BP-1 (TeGAPDH/CP12/PRK) was solved at an overall resolution of 4 [106]. This research revealed critical details on the ternary complex assembly and the CP12 structure; however, the CP12–PRK interaction surfaces were poorly resolved, with a local resolution of 6.2 Å. For further clarification of the precise intermolecular interactions of the ternary complex GAPDH/CP12/PRK, high-resolution structures are required.

### 2.4. A Chloroplast Purinosome

The purinosome is a multi-enzyme complex that carries out *de novo* purine biosynthesis within the cell [107]. Six identified enzymes were postulated to directly participate in a ten-step biosynthetic pathway converting phosphoribosyl pyrophosphate to inosine monophosphate [108]. In human cells, the *de novo* purine biosynthesis pathway has been proved to form a multi-enzyme complex to facilitate substrate channeling between each enzyme of the pathway [108,109,110]. Remarkably, under cellular circumstances that lead to a high demand for purines, the enzymes of *de novo* purine biosynthesis cluster together into multi-enzyme complexes that have been dubbed purinosomes [109] (Figure 4B). Proteomics has subsequently been used to define the nature of protein–protein interactions within the cluster demonstrating that a core complex which assembles in a step-wise fashion includes the first three enzymes in the pathway. Subsequent study of the mutants of other enzymes in the pathway revealed a decreased purinosome formation suggesting that these proteins likely affect complex stability [111]. In one of the recent studies, metabolomics and state-of-the-art gas cluster beam secondary ion mass spectrometry (GCIB-SIMS) were used to directly visualize *de novo* purine biosynthesis in the enzyme–enzyme assembly known as the purinosome (Pareek et al., 2020). In plants, the purine nucleotide enzymes are localized to the chloroplast raising the possibility of a chloroplast purinosome. However, the association of this complex in plants yet not to be demonstrated.

The previous sections have defined well characterized and or putative supramolecular assemblies. In the rest of the article, we will define methods to identify novel super-complexes and to assess the possibility that they mediate substrate channeling.

## 3. Methods to Identify Super-Complexes

Size-exclusion chromatography (SEC) or gel filtration are well established techniques for the super-complex purification relying on the ability, or lack thereof, of particles to pass through a column of porous beads according to their hydrodynamic radius. Recently, the conserved protein complexes of 13 plant species were identified by using size exclusion chromatography with mass spectrometry [112,113]. One of the most significant advantages of this technology is that it may be used under physiological conditions, allowing researchers to study the dynamics of protein complex formation and dissociation. For specific enrichment of super-complexes, the expression and purification recombinant proteins fused with tags can also be enriched by affinity purification. After digestion of recombinant proteins to remove the tags from mature proteins, gel filtration chromatography was applied to isolate free proteins. Each purified protein was checked by SDS-PAGE and identified by mass spectrometry. Furthermore, because of the danger of subunit losses, numerous purification procedures are not always an option. Sucrose density gradient ultracentrifugation and blue-native polyacrylamide gel electrophoresis (BN-PAGE) are two typical methods for purifying super-complexes. The latter approach does not guarantee the highest level of protein purity.

BN-PAGE is the most suitable strategy to separate super-complexes of the OXPHOS system and was first introduced by Schägger et al. [104] BN-PAGE was usually used identify the largest stable protein complexes which can withstand solubilization and large-pore acrylamide gels was used to separate super-complexes and patches of larger such complexes of 10 MDa [105]. In addition, sucrose density gradient centrifugation was used to purify a range of PSII super-complexes with different antenna compositions [106]. Both approaches, however, take around half a day or more to properly separate the largest super-complexes, which may be inconvenient for the most labile. The rarely used approach of free-flow electrophoresis (FFE) could be a viable alternative. FFE has the advantage of not requiring any interaction with a solid matrix, as is the case with BN-PAGE. The short time between solubilization and particle isolation is another significant benefit of FFE separations (1 h or less). More research is needed to optimize the usage of FFE for big particle separation (quantity and resolution for the particles, standardization plus mode of operation for the instrument). Because FFE separates particles solely on the basis of their charge, it will undoubtedly open up new avenues for the creation of unique super-complexes. For a long time, X-ray diffraction was used to determine the structure of the big protein complex, but single particle electron microscopy has lately emerged as a viable alternative due to improved microscopes and image detectors.

## 4. Substrate Channeling of the Super-Complex

The substrate channeling is the super enzyme complex assembled by transient and protein-surface structure dependent interaction in order to passing of the intermediary metabolic products, and thereby regulate metabolic flux by association and dissociation of the components [12,114]. Various benefits of substrate channeling are suggested such as local enrichment of metabolite concentrations in order to achieve high reaction rates, isolation of intermediates from competing reactions, sequestration of cytotoxic metabolites and protection of unstable intermediates [114,115,116]. The “metabolon” was first defined in 1985 by Paul Srere as “a supramolecular complex of sequential metabolic enzymes and cellular structural elements” [91].

Srere also suggested that the enzyme super-complex is assembled by transient, protein-surface, structure-dependent interactions that maintain substrate channeling, whereby the intermediates are maintained within the metabolon and the interconversions are catalyzed by sequential enzymes [117]. In addition, the amount of functional or physiological evidence available in support of channeling between the super-complexes is even less than that from structural studies [118], for instance, the substrate channelings of PSI–LHCI and the respiratory has been proposed for the long term based on the protein structure of the super-complex. In plants, only the pathways of glycolysis [119], the TCA cycle [120], the upper pathway of phenylpropanoid biosynthesis [121] and the cyanogenic glucoside biosynthetic pathway [122] evaluated both protein interaction and substrate channeling to prove the metabolons [110,118].

## 5. Outlook

The functional significance of protein super-complexes is not entirely clear because of the transient and dynamical association. The dynamic association of the plastic super-complexes may also mediate the substrate channel to improve the efficiency of the flux. For example, the organization of PSI–LHCI super-complexes was suggested to mediate the substrate channeling which could efficiently transfer energy absorbed by LHCI to the PSI core by avoiding concentration quenching. However, it is still technically difficult to monitor these assembly in vivo. More evidence on the super-complexes association and the physiological function will be obtained from the cryo-electron microscopy.

## Figures and Tables

**Figure 1 biomolecules-11-01839-f001:**
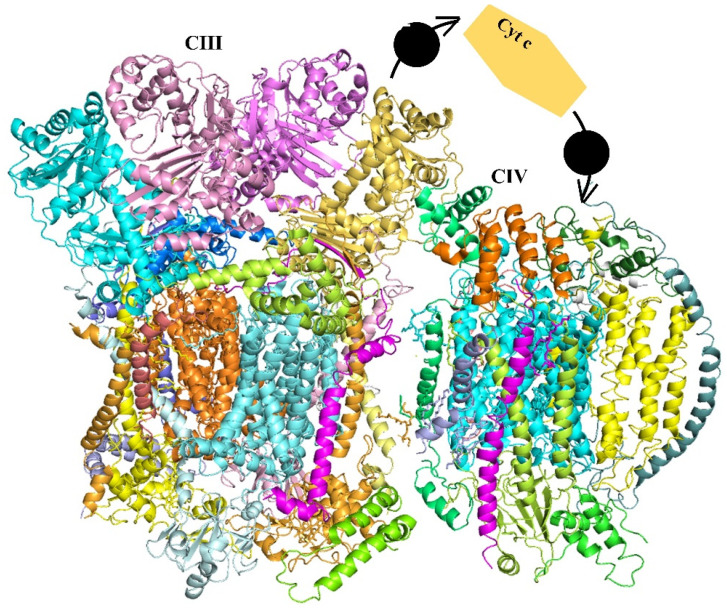
Structure of the respiratory chain super-complex. cryo-EM maps of respiratory chain complex from *Saccharomyces cerevisiae*. (resolution 3.17 Å).

**Figure 2 biomolecules-11-01839-f002:**
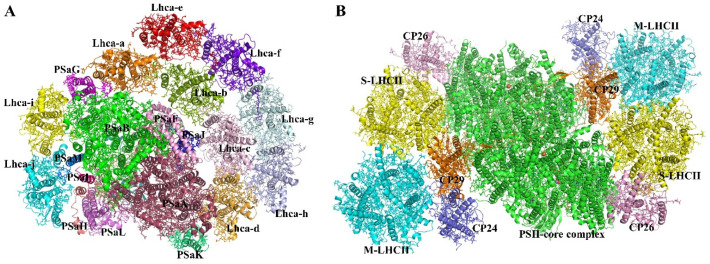
Structure of the PSI–LHCI and PSII–LHCII super-complex. (**A**) cryo-EM maps of super-complexes with PSI and LHCI from *Bryopsis corticulans*. (resolution 3.49 Å). PSaA, raspberry; PSaB, green; PSaF, pink; PSaG, magenta; PSaH, salmon; PSaI, hot pink; PSaJ, blue; PSaK, lime green; PSaL, violet; PSaM, marine; Lhca-a, orange; Lhca-b, split pea; Lhca-c, light pink; Lhca-d, bright orange; Lhca-e, red; Lhca-f, purple blue; Lhca-g, pale cyan; Lhca-h, light blue; Lhca-i, yellow; Lhca-j, cyan. (**B**) cryo-EM maps of super-complexes with PSII core and LHCII from *Pisum sativum*. (resolution 2.70 Å). M–LHCII, cyan; S–LHCII, yellow; CP24, slate; CP29, orange; CP26, light pink; PSII core complex, green.

**Figure 3 biomolecules-11-01839-f003:**
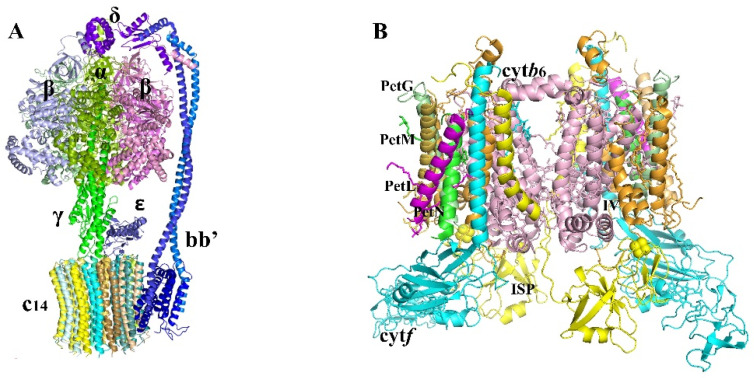
Structure of the ATPase, cyto*b6f*. (**A**) cryo-EM maps of ATPase from *Spinacia oleracea*. (resolution 3.35 Å). α, limon, light pink and light blue; β, magenta, pale green and split pea; γ, green; ε, slate; δ,purple blue; c14, yellow and cyan gradual part; bb’, blue and marine. (**B**) cryo-EM maps of super-complexes with cyto*b6* and cyto*f* from *Spinacia oleracea*. (resolution 3.58 Å). cyto*b6*, light pink; cyto*f*, cyan; ISP, yellow; IV, orange; PetG, pale green; PetM, wheat; PetL, magenta; PetN, green.

**Figure 4 biomolecules-11-01839-f004:**
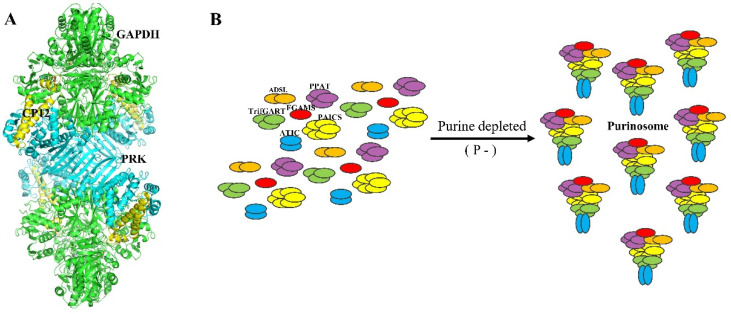
Structure of the GAPDH-CP12-PRK complex and the purinosome formation. (**A**) cryo-EM maps of complexes with GAPDH, CP12 and PRK from *Thermosynechococcus vestitus*. (resolution 3.90 Å). GAPDH, green; CP12, yellow; PRK, cyan. (**B**) Enzymes of *de novo* purine biosynthesis assemble into purinosome in purine depleted conditions in Hela cells. PPAT: amidophosphoribosyl transferase, purple; FGAMS: phoshoribosylformylglycinimidine transferase, red; PAICS: bifunctional phosphoribosylaminoimidazole carboxylase and phosphoribosyl aminoimidazole (CAIR synthetase, and SAICAR synthetase), yellow; ATIC: bifunctional 5-aminoimidazole-4-carboxamide nucleotide formyltransferase/IMP cyclohydroxylase (IMPCH), blue; ADSL: bifunctional adenylosuccinate lyase, orange; GART: trifunctional phosphoribosylglycinamide formyltransferase (GAR synthetase, GAR formyltransferase, and AIR synthetase), green.

## Data Availability

Not applicable.

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
