# Peer review of "The Assembly of Super-Complexes in the Plant Chloroplast"

_biomolecules, 2021, doi:10.3390/biom11121839_

Round 1

Reviewer 1 Report

This is an interesting comprehensive review on super complexes in the chloroplast. I have only one major comment: The first paragraph about LHCs on page 3 is redundant with the detailed descriptions given later. This part is hard to follow for a non-expert. It should be removed or parts of it may be shifted to a place behind the detailed description of PSI and PSII.

In addition, it may be of interest to add a chapter (even if it is highly speculative) on the factors that may drive assembly and disassembly of super complexes under different physiological conditions (lipid composition expression of special proteins, ion concentrations, posttranslational modifications etc).

Minor comments:

p.1, l. 33 ; I guess it should be “substrate channeling or proton translocation” If not, it should be explained what this means.

p.3, 1st para. If you decide to keep this part, it should be clearly said that NPQ that takes place in the antenna of PSII (LHCII)

p.3, l. 128, what is an Lhcr protein?

p.4, l. 150, this is wrong. 1 pheophytin is bound to DA and the second to D2.

p.5, can you explain what is an extensive interaction compared to a hydrophobic or electrostatic interaction? This term is not clear.

p.7, l. 331, beta-carotene

p.7, l. 329, is 18 years ago still recently?

p.7, l. 333, missing link: introduce cyclic flow here or the putative role of the heme ci in cyclic?

p.7, l. 337, proton uptake in the stroma and proton release to the lumen

p.8, l. 349, stroma lamellae

p.9, l. 382-385, it’s the proton gradient that protects against photoinhibition independent on the way it is generated. It don’t understand your argument. The NDH complex in higher plants (not in green algae), pumps protons very efficiently like complex I. The flow may be low because it is a minor complex but it contributes nevertheless to the proton gradient and thereby also to NPQ

p.8, l. 344 and p. 9, l. 395. The two statements are contradictory

Author Response

Review1:

Comments and Suggestions for Authors:

This is an interesting comprehensive review on super complexes in the chloroplast. I have only one major comment: The first paragraph about LHCs on page 3 is redundant with the detailed descriptions given later. This part is hard to follow for a non-expert. It should be removed or parts of it may be shifted to a place behind the detailed description of PSI and PSII.

Response: Thank you for your suggestions, we removed this part and shifted some parts to the description of PSI.

In addition, it may be of interest to add a chapter (even if it is highly speculative) on the factors that may drive assembly and disassembly of super complexes under different physiological conditions (lipid composition expression of special proteins, ion concentrations, posttranslational modifications etc).

Response:   There is very little researches on the assembly and disassembly of plastic supercoomplexes.

We have however added few sentences at the outlook.

Minor comments:

p.1, l. 33 ; I guess it should be “substrate channeling or proton translocation” If not, it should be explained what this means.

Response:  It’s not “or”. We think in the mETC complex, there might be substrate channeling existing between the proteins to transport protons from one to another reaction core, which works as the metabolon.

p.3, 1st para. If you decide to keep this part, it should be clearly said that NPQ that takes place in the antenna of PSII (LHCII)

Response: we deleted this part.

p.3, l. 128, what is an Lhcr protein?

Response: It should be “Lhcb protein”.

p.4, l. 150, this is wrong. 1 pheophytin is bound to DA and the second to D2.

Response: Thanks for your correction. We have updated it.

p.5, can you explain what is an extensive interaction compared to a hydrophobic or electrostatic interaction? This term is not clear.

Response: we added P.5, L.185-187 “The contacts between M-LHCII and S-LHCII′ or M-LHCII and CP29, on the other hand, are less extensive and mostly consist of hydrophobic interaction, in dependent of the enthalpy and entropy rather than intermoleculars forces such as der Waals forces or hydrogen bonds.”

p.7, l. 331, beta-carotene

Response: updated

p.7, l. 329, is 18 years ago still recently?

Response: we correct it with specific year “in 2003”.

p.7, l. 333, missing link: introduce cyclic flow here or the putative role of the heme ci in cyclic?

Response: we added P.7, L.317-319. “This heme structural information suggests it is an essential factor for quinone reduction at the Qi site and also facilitate rapid electron transfer and significant interactions between hemes ci and bi

p.7, l. 337, proton uptake in the stroma and proton release to the lumen

Response: updated

p.8, l. 349, stroma lamellae

Response: updated

p.9, l. 382-385, it’s the proton gradient that protects against photoinhibition independent on the way it is generated. It don’t understand your argument. The NDH complex in higher plants (not in green algae), pumps protons very efficiently like complex I. The flow may be low because it is a minor complex but it contributes nevertheless to the proton gradient and thereby also to NPQ

Response: Based on the paper results, NPQ induction was not affected in the NDH activity impaired Arabidopsis. And tobacco ndh knockout lines sensitive to some environmental stress, suggesting that NDH-dependent cyclic flow is essential in photoprotection.

Munekage, Y.; Hashimoto, M.; Miyake, C.; Tomizawa, K.-I.; Endo, T.; Tasaka, M.; Shikanai, T. Cyclic electron flow around photosystem i is essential for photosynthesis. Nature 2004, 429, 579-582.

p.8, l. 344 and p. 9, l. 395. The two statements are contradictory

Response: We think they are not contradictory. In the paragraph 8, we stated that cytb6f interacted with PSI forming a super-complex. And in the paragraph 9, we mentioned that “PSI–LHCI–LHCII–FNR–cyt b6/f–PGRL1 super-complex” was still about core complex of cytb6f and PSI, so actually they are the same.

Reviewer 2 Report

The review is well written and contains quite a bit of information. There are a few minor points that could aid in the clarity of the manuscript;

a) The figure legends start with ' Crystal structure of respiratory chain supercomplex...' But most of these are not crystal structures but  Cryo EM structures. The figure legends need to be edited.

b) In the ' methods to identify super-complex section, the authors would want to add information about possible methods for specific enrichment of supercomplexes for structural Biology applications. Are there affinity purification methods using antibodies or subunit proteins available?

c) The abstract and the manuscript itself requires extensive English language editing. The overall quality of narration needs to be improved.

Some examples of typos which needs to be corrected 

Line 530 : 'supercomplexes may mediated the substrate'

Line 19: 'Here, we 19 will detail discuss core complexes '

Author Response

Review2:

Comments and Suggestions for Authors:

The field of supercomplexes is very strongly debated for mitochondrial respiratory chain, but is rather scant for chloplast electron transfer. The authors fill the gap with this updated and important review. I find the review well organized and readible. It is important that the authors give reference to similar aspects in mitochondria. Perhaps the reference to possible channeling coud be esomewhat extended.

A minor corrrection: at line 31, Complex II is not usually considered part of a supercomplex and does not pump protons.

Response: We have modified this statement accordingly.

Reviewer 3 Report

The field of supercomplexes is very strongly debated for mitochondrial respiratory chain, but is rather scant for chloplast electron transfer. The authors fill the gap with this updated and important review. I find the review well organized and readible. It is important that the authors give reference to similar aspects in mitochondria. Perhaps the reference to possible channeling coud be esomewhat extended..

A minor corrrection: at line 31, Complex II is not usually considered part of a supercomplex and does not pump protons.

Author Response

Review3:

Comments and Suggestions for Authors:

The review is well written and contains quite a bit of information. There are a few minor points that could aid in the clarity of the manuscript;

  1. a) The figure legends start with ' Crystal structure of respiratory chain supercomplex...' But most of these are not crystal structures but Cryo EM structures. The figure legends need to be edited.

Response: we edited it to state “Structure…” instead.

  1. b) In the ' methods to identify super-complex section, the authors would want to add information about possible methods for specific enrichment of supercomplexes for structural Biology applications. Are there affinity purification methods using antibodies or subunit proteins available?

Response: Thanks for your suggestion. We added in L.467-471. “For specific enrichment of super-complexes, the expression and purification recombinant proteins fused with tags can also be enriched by affinity purification. After digestion of recombinant proteins to remove the tags from mature proteins, gel filtration chromatography was applied to isolate free proteins. Each purified proteins was checked by SDS-PAGE and identified by mass spectrometry.”

  1. c) The abstract and the manuscript itself requires extensive English language editing. The overall quality of narration needs to be improved.

Response: We have improved it. However, please note one of the author is a native speaker who has published several hundred articles.

Some examples of typos which needs to be corrected 

Line 530: 'supercomplexes may mediated the substrate'

Line 19: 'Here, we 19 will detail discuss core complexes'

Response: corrected.